# Trends of Peatland Research Based on Topic Modeling: Toward Sustainable Management under Climate Change

Hyunyoung Yang [1], Jeongyeon Chae [1], A-Ram Yang [1], Rujito Agus Suwignyo [2] and Eunho Choi [1,*]

1   Global Forestry Division, Future Forest Strategy Department, National Institute of Forest Science, Seoul 02455, Republic of Korea; hyhy0672@korea.kr (H.Y.); jyuna5@naver.com (J.C.); aryang@korea.kr (A.-R.Y.)
2   Department of Agronomy, Faculty of Agriculture, Universitas Sriwijaya, Palembang 30662, Indonesia; rujito@unsri.ac.id
*   Correspondence: ehchoi710@korea.kr; Tel.: +82-2-961-2882

**Abstract:** Peatlands are wetlands with an accumulation of peats, partially decomposed organisms, under waterlogged and anoxic conditions. Despite peatlands being extensively studied due to their wide distribution and various functions, the trends in peatland research have hardly been analyzed. We performed dynamic topic modeling (DTM) and network analysis to investigate the changes in the global trends in peatland research. Among the searched studies using the keyword 'peatland' from ScienceDirect, titles and abstracts from 9541 studies (1995–2022) were used for the analysis. They were classified into 16 topics via DTM (geomorphology, land use and land cover, production, greenhouse gas, habitat, permafrost, management, deposit, fire, soil organic matter, peatland formation, forest, past environmental change, microbe, metal, and hydrology). Among these, the proportion of 'management' was the largest and increased the fastest, showing the transition of research trends toward the sustainable management of peatlands under climate change. The keywords used within topics tended to change dynamically when related to a large number of studies and increasing trends. Network analysis among topics suggested that studying peatlands as a response measure to climate change will promote overall peatland research because the greenhouse gases topic had the greatest impact on other topics. Despite increasing research on peatland management under climate change, a gap between academia and policies was found in the field of using peatlands as a response measure to climate change, indicating the necessity for effective policies, research, and technology. This study demonstrates that DTM and network analysis are useful tools for understanding the temporal shift of views on peatlands and finding a gap we need to focus on in the near future.

**Keywords:** peatland; dynamic topic modeling; network analysis; research trend analysis




## 1. Introduction

Peatlands are wetland ecosystems with an accumulation of peat, which is partially decomposed organic matter from dead plants and animals, under waterlogged and anoxic conditions [1,2]. Peatlands can be defined in several ways; however, the most common definition refers to peatlands as follows: (1) where more than a 30 cm depth of peat has accumulated or (2) where dead organic material comprises at least 30% of the dry mass of the peat [3,4].

Peatlands are distributed across broad areas in tropical, temperate, and subarctic regions and are characterized by climate-dependent distinctions. The global area of peatland was estimated as 4.232 M km$^2$, with 75% located in the northern area (>30° N) and the remaining located in the tropical area (30° S to 30° N) [5,6]. Tropical peatlands have rapid production and decomposition rates due to high temperatures and heavy rainfall; however, the decomposition rate is relatively slower than the production rate due to the

high lignin and cellulose contents of trees, resulting in the accumulation of peat [7,8]. In temperate peatlands with relatively lower rainfall and temperatures than tropical regions, peat is accumulated by the delayed decomposition of herbaceous plants, such as reeds or sedges, in inundated anaerobic environments. In cold and humid subarctic regions, peat is generated due to the delayed decomposition rate of moss in inundated environments [9].

Peatlands provide important ecological, economic, and cultural roles. For instance, they can regulate regional and global climate by storing more than 600 gigatons of carbon, which represents up to 44% of global soil carbon [10], despite covering less than 3% of the global land surface [5]. These ecosystems can also store water by regulating the river's discharge [11] and improving the water cycle [12]. Even though the nutrient availability is low, peatlands can store nitrogen in the ecosystem [13]. Moreover, they also have an important role in conserving biodiversity, which supports the ecosystem processes and services [14,15]. Peatlands also provide economic benefits with livelihoods for local communities, such as agricultural cultivation and fishing [16,17]. Furthermore, healthy peatlands are repositories of cultural and social value relating to heritage, education, sense of place, recreation, and spirituality [18].

Although peatlands provide such diverse benefits, they are fragile and subject to serious damage. The main causes of degrading peatlands are deforestation, drainage, fire, and conversion to other land types such as agriculture. This results in increased greenhouse gas emissions to the atmosphere, reduced carbon storage, enhanced risk of flooding, changes in nutrient storage and cycling, and biodiversity loss [17,19]. Furthermore, smoke haze due to the peatland fires is not only a serious public health concern but also a cause of conflicts between neighboring countries [20,21]. As a caution against the crises that peatlands face deepens, global efforts to restore peatlands have been also growing [17,22,23].

Despite peatlands being extensively studied due to their diverse values, the crises they face, and the efforts to prevent and solve the problems, there is limited analysis of the trends in peatland research. In the only previous study, Van Bellen and Larivière [24] collected papers published from 1991 to 2017 under the keywords 'peatland', 'bog', 'fen', and 'mire' in Web of Science to examine the global trend in peatland research. Based on the keywords assigned for each paper by Web of Science, they considered the top 20 of the most frequent keywords as their research topics and described how each topic changed over time. This study was a first attempt to understand the trends in peatland research based on the frequency of keywords; however, the criteria for keyword allocation to each paper were unclear, and the keyword itself was considered one topic, resulting in the issue of overlapping topics. In addition, their approach could not ascertain the structural relationships among topics due to the overlapping issue. To overcome these limitations, several keywords with a high probability of appearing on a specific theme should be grouped as a single topic, and the relationships among topics should be analyzed using an improved methodology such as topic modeling and network analysis.

Topic modeling is a text mining technique that helps to find hidden semantics in document collections and cluster them into topics. The most commonly used topic modeling technique is Latent Dirichlet Allocation (LDA), which is based on the probabilistic algorithm of unsupervised learning [25,26]. However, LDA is a static model that does not account for topic evolution over time. To improve this, dynamic topic modeling (DTM) was proposed to analyze the time evolution of topics in large document collections [27]. Dynamic topic modeling can identify the temporal change in keyword occurrence probability within a topic; therefore, it has been applied across various fields to analyze research trends (e.g., Sha et al. [28]; Yao & Wang [29]). In addition, DTM is often used with network analysis to analyze the structural relationships among topics [30,31].

Using DTM and network analysis, this present study attempted to answer the following question, 'How have the global trends in peatland research changed over time?' To answer this question, we clustered topics, analyzed the temporal change in topic proportions and keyword occurrence probabilities within topics based on DTM, and conducted a network analysis to reveal the structural relationships among topics in order to provide a

comprehensive understanding of global peatland research. Knowing the research trend of peatlands can help us understand the temporal shift of views on peatlands and find a gap we need to focus on in the near future.

## 2. Materials and Methods

### 2.1. Research Framework

Figure 1 shows a diagram of the research procedures. From the database of ScienceDirect, the titles and abstracts of papers published until 2022 were collected by searching for the keyword 'peatland'. After data pre-processing, we determined the optimal number of topics based on coherence score. Then, we applied DTM to titles and abstracts of papers published from 1995 to 2022, when all of the topics began to appear. To comprehend the clustered topics, suitable topic names were chosen by the researchers based on the keywords extracted from the papers and representative papers with a high probability of being allocated to each topic. Subsequently, the temporal changes in the proportion of each topic and the temporal change in the probability of keywords within topics were analyzed. Network analysis was also performed to analyze the structural relationships among topics.

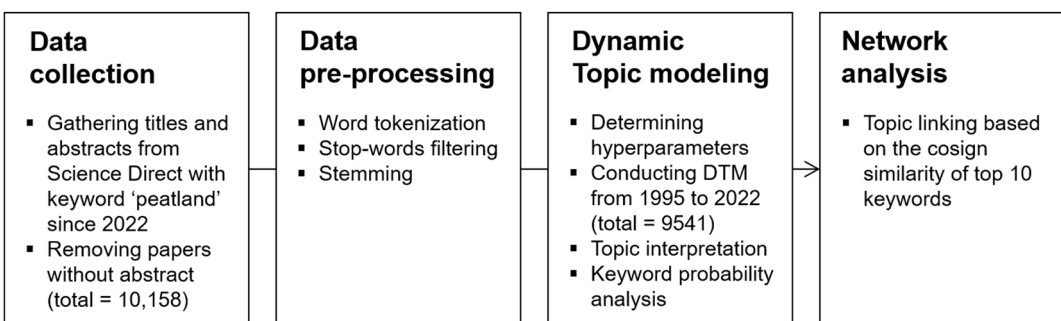

**Figure 1.** Diagram of the research methods followed to investigate changes in trends of peatland research.

### 2.2. Data Collection and Pre-Processing

We searched 11,519 English-written papers published until 2022, using the keyword 'peatland' on the global academic publication data platform ScienceDirect. We did not limit the search range to title, abstract, and author keywords, but as whole document for two reasons: (1) If 'peatland' is searched limited to title, abstract, and author keywords, many papers that actually used peatland as a study area but mentioned this term only in Materials and Methods or other sections were excluded. It can be a critical limitation in analyzing global trends of peatland research. (2) Even if some papers did not study peatland directly, we believe that it was necessary to include papers using 'peatland' in the papers as a comparison target or mentioning 'peatland' as a solution to climate change mitigation for understanding the comprehensive research trend of peatland. Thus, we adopted a method that can include as many papers related to peatland as possible. Among the collected papers, 1361 papers that did not include abstracts or were indicated as 'unknown' were excluded as outliers. Finally, 10,158 papers were found from 1953 to 2022. To pre-process the collected data, tokenization was performed first to divide the text data into tokens, which are the minimum units of meaning. Since not all words that underwent tokenization were meaningful, we eliminated elements such as English stop-words, special characters, and punctuation marks provided in the Natural Language Toolkit of Python (3.9.12 ver.). Additionally, verbs, adverbs, and adjectives frequently used in all studies were also eliminated (Table S1). Words of less than three characters were eliminated, and stemming and lower-case conversion was performed for all words using Python to prevent the repetition of singular and plural words.

### 2.3. Determination of Hyperparameters

In DTM, the number of topics is the most important hyperparameter. Selecting an insufficient number of topics will make the content of one topic too broad, while selecting

too many topics will result in smaller and fairly similar topics. There are criteria used to determine the optimal number of topics, such as perplexity, log-likelihood, and coherence score [26]. This study determined the number of topics based on the commonly used criteria, coherence score [25,29,32]. The coherence score is an index that measures the coherence of the keywords constituting a topic, and a higher coherence score indicates a better description of the topic by the keywords [33]. In order to find the optimal frequency of DTM training (i.e., passes) and the number of topics, the modeling was performed several times while adjusting the variables, and the coherence score was the highest at 0.059 when the number of topics was 16 with 40 passes (Figure 2).

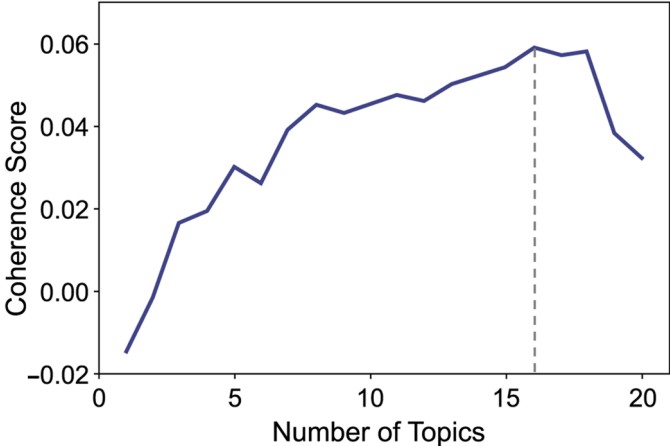

**Figure 2.** Coherence score of dynamic topic modeling depends on the number of topics. The highest coherence score was observed for 16 topics marked by a dash line.

### 2.4. Dynamic Topic Modeling

Topic models are unsupervised machine learning algorithms to identify themes in large text collections. They assign the document with different weights to the topics and allocate the words based on two distributions (i.e., the distribution of words in the topics and the distribution of topics in the documents). Among topic models, DTM considers the temporal evolution of topics, assuming that there is $K$ number of topics over the entire period and $K_{t-1}$ of time $t-1$ evolves to topic $K_t$ of time $t$. It means each time $t$ should contain the $K$ number of topics. This method can analyze the temporal change in the probability distribution of keywords representing each topic [31,34].

Among the collected data, studies published from 1995 to 2022, the period when all 16 topics appeared simultaneously, were used for DTM analysis. Dynamic topic modeling was performed using the 'ldaseqmodel' library in Python, and papers and keywords were classified into each topic according to the optimal number of topics. While the computer classifies each topic, the content must be interpreted by researchers. Therefore, two researchers individually interpreted the topics based on the top keywords and representative papers (Table S2) that had a high probability of being assigned to each topic. The topics were reviewed and corrected by all co-authors. Subsequently, linear regression analysis was performed to test the significance of temporal tendency in the number of papers for each topic (i.e., topic frequency) and the ratio of the number of papers in a specific topic to the total number of papers (i.e., topic proportion). The year of publication was set as an independent variable, and topic frequency and topic proportion by year were set as dependent variables. Using Python's 'statsmodels' library, the increase or decrease in each topic over time was analyzed. The significance level was set to 95%. When the regression coefficient was positive, the topic was classified as a hot topic (i.e., significantly increases with time); when the regression coefficient was negative, it was classified as a cold topic (i.e., significantly decreases with time). The temporal change in the keyword occurrence probability within the topics was also analyzed based on the top five keywords.

### 2.5. Network Analysis

Network analysis can describe the relationships between the study targets in a combination of nodes (i.e., points) and links (i.e., lines). Important nodes can be decided based on the extent of connection between nodes. The degree of node importance is expressed as centrality, which means the degree to which a particular node is positioned at the center of a network while having concentrated connections to other nodes [35,36]. This study calculated degree centrality, closeness centrality, and eigenvector centrality. The degree of centrality increases as the number of links directly connected to a certain node increases, representing local centrality. The closeness centrality increases as the sum of the shortest distances from a certain node to other nodes decreases, representing the global centrality across the entire network [37,38]. Meanwhile, eigenvector centrality is an extended concept of degree centrality, considering not only the number of nodes directly connected to a certain node but also the centralities of the other nodes, reflected as weights. Thus, the eigenvector centrality increases as it connects to a node with high centrality, and it is useful in identifying the node with the largest impact on the entire network [36,39]. This study used the 'networkx' library in Python to analyze the network among topics. Each topic was set as a node, and the cosine similarity distance among topics was set as the link to calculate the three centrality types. The cosign similarity distance was calculated using the cosine similarity value for the term frequency-inverse document frequency (TF-IDF) matrix of the top keywords of each topic. Here, TF-IDF is a traditional statistics-based text similarity measure algorithm that constructs model using text word frequency vector [40].

## 3. Results

### 3.1. Dynamic Topic Modeling

#### 3.1.1. Topic Interpretation

Figure 3 shows the number of peatland papers published in ScienceDirect from 1953 to 2022. Before 1980, the number of papers was negligible (less than 1%); however, it increased exponentially thereafter. During the DTM analysis period from 1995 to 2022, the number of papers was 9541, accounting for 94% of the total papers.

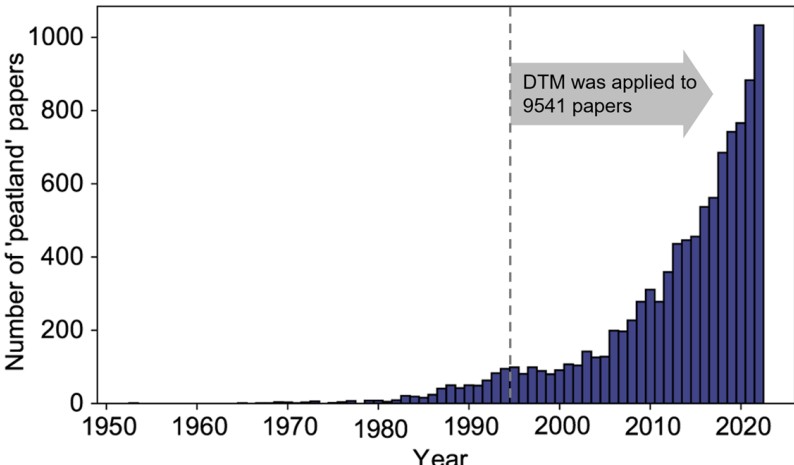

**Figure 3.** Number of papers published between 1953 and 2022 that were identified using the keyword 'peatland' on ScienceDirect.

The DTM results showed that peatland research was classified into 16 topics (Table 1). The probability of a specific keyword appearing in each topic was calculated every year and summed during the entire period. The keywords are presented in descending order of the sum of these values in Table 1. The topics were named based on the top keywords and representative papers of each topic.

**Table 1.** Top keywords with high probability appearing in each topic calculated using dynamic topic modeling in peatland research.

| No. | Topic 1 | Topic 2 | Topic 3 | Topic 4 | Topic 5 | Topic 6 | Topic 7 | Topic 8 |
| --- | --- | --- | --- | --- | --- | --- | --- | --- |
| | Geomorphology | Land Use and Land Cover | Production | Greenhouse Gas | Habitat | Permafrost | Management | Deposit |
| 1 | sediment | use | energy | soil | restor(ation) | surfac(e) | ecosystem | coal |
| 2 | deposit | base | product | carbon | agricultur(e) | water | servic(e) | deposit |
| 3 | glaci(er) | model | use | emiss(ion) | conserv(ation) | groundwat(er) | base | alkan |
| 4 | marin(e) | area | fuel | flux | habitat | thaw | sustain(ability) | basin |
| 5 | eros(ion) | variabl(e) | wast(e) | ecosystem | land | climat(e) | global | plant |
| 6 | coastal | veget(ation) | treatment | warm | area | region | manag(ement) | format(ion) |
| 7 | river | spatial | biomass | global | farm | temperatur(e) | polic(y) | isotop(e) |
| 8 | environ(ment) | map | remov(e) | nitrogen | palm | layer | review | peat |
| 9 | channel | result | biofuel | greenhous(e) | speci(es) | hydrolog(y) | environ(ment) | valu(e) |
| 10 | system | resolut(ion) | materi(al) | increas(ing) | landscap(e) | arctic | land | lignit(e) |
| 11 | fluvial | time | system | temperatur(e) | natur(e) | flow | econom(y) | seam |
| 12 | form | classif(ication) | industri(al) | cycl(e) | biodivers(sity) | soil | resourc(e) | condit(ion) |
| 13 | area | monitor(ing) | base | rate | cover | condit(ion) | develop(ment) | environ(ment) |
| 14 | valley | scale | biochar | product | plant | snow | mitig(ation) | composit(ion) |
| 15 | lake | approach | peat | effect | increase(ing) | zone | climat(e) | domin(ate) |

| No. | Topic 9 | Topic 10 | Topic 11 | Topic 12 | Topic 13 | Topic 14 | Topic 15 | Topic 16 |
| --- | --- | --- | --- | --- | --- | --- | --- | --- |
| | Fire | Soil organic matter | Peatland formation | Forest | Past environmental change | Microbe | Metal | Hydrology |
| 1 | fire | soil | peat | forest | climat(e) | commun(ity) | metal | wetland |
| 2 | atmospher(e) | organ(ic) | peatland | boreal | region | speci(es) | concentr(ation) | water |
| 3 | burn | matter | water | plant | record | microbi(al) | mercuri(al) | river |
| 4 | pollut(ion) | concentr(ation) | depth | stand | holocen(e) | divers(ity) | element | lake |
| 5 | sourc(e) | miner(al) | restor(ation) | increas(ing) | lake | abund(ance) | organ(ic) | stream |
| 6 | deposit | properti(es) | carbon | tree | reconstruct(ion) | plant | MeHg | hydrolog(y) |
| 7 | wildfir(e) | content | tabl(e) | speci(es) | increas(ing) | soil | contamin(ation) | flow |
| 8 | region | condit(ion) | moss | disturb(ance) | temperatur(e) | composit(ion) | iron | area |
| 9 | concentr(ation) | activ(e) | drain | site | chang(e) | bacteri(a) | trace | flood |
| 10 | emiss(ion) | composit(ion) | accumul(ation) | nutrient | china | structur(e) | humic | coastal |
| 11 | anthropogen(ic) | sampl(e) | drainag(e) | harvest | mountain | ecolog(y) | water | aquat(ic) |
| 12 | activ(e) | decomposit(ion) | surfac(e) | pine | veget(ation) | group | natur(e) | catchment |
| 13 | area | chemic(al) | condit(ion) | litter | past | testat(e) | compound | load |
| 14 | mine | carbon | increas(ing) | root | core | function | dissolv(e) | qualit(y) |
| 15 | aerosol | acid | site | effect | pollen | environ(ment) | sediment | concentr(ation) |

Only the keywords extracted from the stem were noted and the full term is completed in parentheses for the integrity of the meaning.

The interpretations for each topic are as follows: Topic 1 is 'geomorphology', which covers studies on the geomorphological characteristics such as formation, development, and erosion processes of peatlands in coasts, glaciers, rivers, and basins. Topic 2 is 'land use & land cover', which is focused on studies that identify and classify the types of land use and land cover, including peatlands, using remote sensing. Topic 3 is 'production', which includes research on energy sources such as oil produced from peatland or other land converted from peatland, development of materials that can replace peat and utilization of peat for wastewater and pollutant treatment. Topic 4 is 'greenhouse gas', which deals with the studies on the exchange of greenhouse gases, such as carbon dioxide ($CO_2$), nitrogen dioxide ($N_2O$), and methane ($CH_4$), between peatland soil and the atmosphere and their controlling environmental factors. Topic 5 is 'habitat', which covers research on habitat selection by various plants and animals inhabiting peatlands, biodiversity, and population restoration. Topic 6 is 'permafrost', which includes research on the surface water and groundwater in peatlands located in permafrost, freezing and thawing, and the role of peatland in preventing permafrost thawing. Topic 7 is 'management', which is focused on the sustainable management of various peatland ecosystem services and their synergy and trade-off relationships. For example, maintaining sustainable peatlands requires balancing the conflicting values of ecosystem conservation, mitigation of greenhouse gas emissions, land use, and local economic development. Topic 8 is 'deposit', which covers research on the composition and classification of deposits in ancient peatlands such as coal, brown coal, and shale. Topic 9 is 'fire', which includes studies on fine dust, smoke, fog, pollution, and carbon emissions caused by human-made or natural fires on peatland. Topic 10 is 'soil organic matter', which mainly covers the quantity, concentration, and characteristics of organic carbon, nitrogen, and other organic matter stored in a peatland soil. Topic 11 is 'peatland formation', which includes studies on the impact of water level, moss, vegetation, and soil water content, contributing to peatland formation and regeneration. Topic 12 is 'forest', which deals with the growth and development, harvest, thinning, and decomposition of trees growing on peatlands. Topic 13 is 'past environmental change', which mainly comprises research that infers the change in climate and vegetation of the Holocene via records of deposits, cores, and pollen in peatlands. Topic 14 is 'microbe', which covers the diversity, classification, and functions of microorganisms such as bacteria, viruses, amoebae, and fungi found in soil or organisms in peatlands. Topic 15 is 'metal', which mainly includes research on heavy metal chemicals accumulated in deposits or organisms on peatlands. For example, studies on methylmercury, lead, copper, and humic acid that easily combine with these heavy metals are included. Topic 16 is 'hydrology', which mainly deals with studies on the hydrological responses, amount of runoff, and concentration of dissolved organic matter in rivers, lakes, basins, and coasts that include peatlands.

Figure 4 summarizes the proportions of each topic during the entire period. Topic 7 (management) accounted for the largest proportion at 15%. Topics 13 (past environmental change) and 4 (greenhouse gas) followed, with approximately 12% each. Conversely, Topic 15 (metal) accounted for the smallest proportion, with less than 2%, followed by Topics 9 (fire), 8 (deposit), 6 (permafrost), and 1 (geomorphology), with less than 3% each.

### 3.1.2. Temporal Change in Topics

The temporal changes in the number of papers on each topic (i.e., topic frequency) and the ratio of the number of papers on a topic to the total number of papers (i.e., topic proportion) were analyzed. For the topic frequencies, all topics were categorized as hot topics (all $p < 0.05$, Table 2). Among them, Topics 7 (management) and 8 (deposit) showed the strongest and weakest rates of increase, respectively.

For the topic proportions, there were hot topics, cold topics, and some topics that were neither hot nor cold (Table 2, Figure 5). There were three hot topics with increasing trends: Topics 7 (management), 13 (past environmental change), and 3 (production). Conversely, there were four cold topics with decreasing trends: Topics 10 (soil organic matter), 12

(forest), 8 (deposit), and 5 (habitat). The others did not significantly increase or decrease over time.

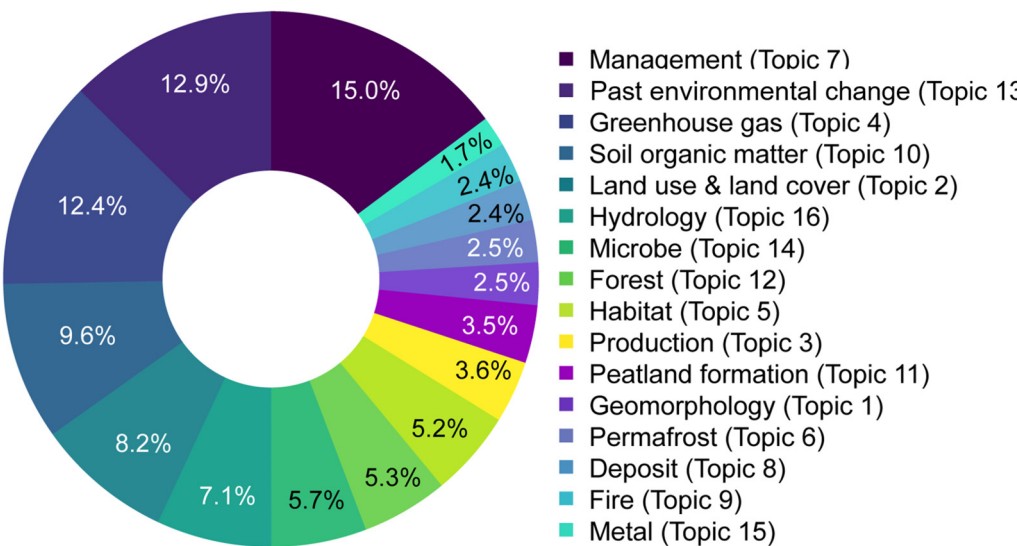

**Figure 4.** Proportions of the 16 topics in peatland research identified via dynamic topic modeling, calculated as the ratio of the number of papers corresponding to a specific topic to the total number of papers from 1995 to 2022.

**Table 2.** Regression coefficients and *p*-values of topic frequency and topic proportion in peatland research from 1995 to 2022.

| | **Frequency** | | | **Proportion** | | |
|---|---|---|---|---|---|---|
| **Topic** | **Coefficient** | ***p*-Value** | **Hot/Cold** | **Coefficient** | ***p*-Value** | **Hot/Cold** |
| Topic 1 | 0.715 | $1.35 \times 10^{-8}$ | Hot | −0.0005 | 0.097 | Normal |
| Topic 2 | 2.576 | $3.19 \times 10^{-13}$ | Hot | 0.0005 | 0.236 | Normal |
| Topic 3 | 1.387 | $2.77 \times 10^{-8}$ | Hot | 0.0009 | 0.013 | Hot |
| Topic 4 | 3.936 | $6.96 \times 10^{-12}$ | Hot | 0.0005 | 0.558 | Normal |
| Topic 5 | 1.401 | $2.88 \times 10^{-8}$ | Hot | −0.0011 | 0.039 | Cold |
| Topic 6 | 0.762 | $2.85 \times 10^{-7}$ | Hot | 0.0002 | 0.928 | Normal |
| Topic 7 | 6.416 | $2.3 \times 10^{-9}$ | Hot | 0.0064 | $3.12 \times 10^{-9}$ | Hot |
| Topic 8 | 0.556 | $1.64 \times 10^{-5}$ | Hot | −0.0014 | 0.006 | Cold |
| Topic 9 | 0.829 | $2.09 \times 10^{-8}$ | Hot | 0.0004 | 0.106 | Normal |
| Topic 10 | 2.144 | $3.75 \times 10^{-9}$ | Hot | −0.005 | $1.41 \times 10^{-6}$ | Cold |
| Topic 11 | 0.850 | $1.58 \times 10^{-11}$ | Hot | −0.0005 | 0.149 | Normal |
| Topic 12 | 0.938 | $1.87 \times 10^{-9}$ | Hot | −0.0028 | $8.86 \times 10^{-5}$ | Cold |
| Topic 13 | 4.241 | $4.47 \times 10^{-13}$ | Hot | 0.0023 | 0.003 | Hot |
| Topic 14 | 1.913 | $1.93 \times 10^{-10}$ | Hot | 0.00052 | 0.307 | Normal |
| Topic 15 | 0.589 | $3.5 \times 10^{-7}$ | Hot | −0.0001 | 0.586 | Normal |
| Topic 16 | 1.761 | $2.8 \times 10^{-12}$ | Hot | −0.0009 | 0.125 | Normal |

### 3.1.3. Temporal Changes in Keyword Occurrence Probability within Topics

To examine the temporal changes in keyword occurrence probability within topics in detail, we selected two topics as examples: (1) Topic 7 (management), which had the largest topic proportion and sharpest increasing rate, and (2) Topic 8 (deposit), which had one of the smallest topic proportions and the weakest increasing rate. The changes in occurrence probability were analyzed for all keywords that appeared within the top five ranking keywords every year. Since the keywords could both climb into the ranking (i.e., appearance) or fall out of the ranks (i.e., disappearance), the number of keywords included in the graph could be more than five.

First, the temporal change in the keyword occurrence probability in Topic 7 (management) was dynamic (Figure 6). The keyword *ecosystem* appeared within the ranking during the entire period, and its probability increased with time. *Servic(e)* began to appear in the ranking in 1999, and its probability also increased with time. Both words consistently maintained their high ranking. In contrast, the probability of the appearance of *develop*, *manag(ement)*, and *environ(ment)* gradually decreased, and they disappeared from the ranking after 1998, 2002, and 2001, respectively. The probability of the appearance of *polic(y)* increased up to 1998, then decreased again and disappeared from the ranking after 2005. *Sustain(ability)* appeared within the ranking in 1999, and this position was consistently maintained up to 2017. *Base*, *global*, and *resourc(e)* were initially not within the ranking; however, they appeared after 2003, 2006, and 2018, respectively. As such, the appearance, disappearance, and change in probability of keywords within the ranking frequently occurred within Topic 7.

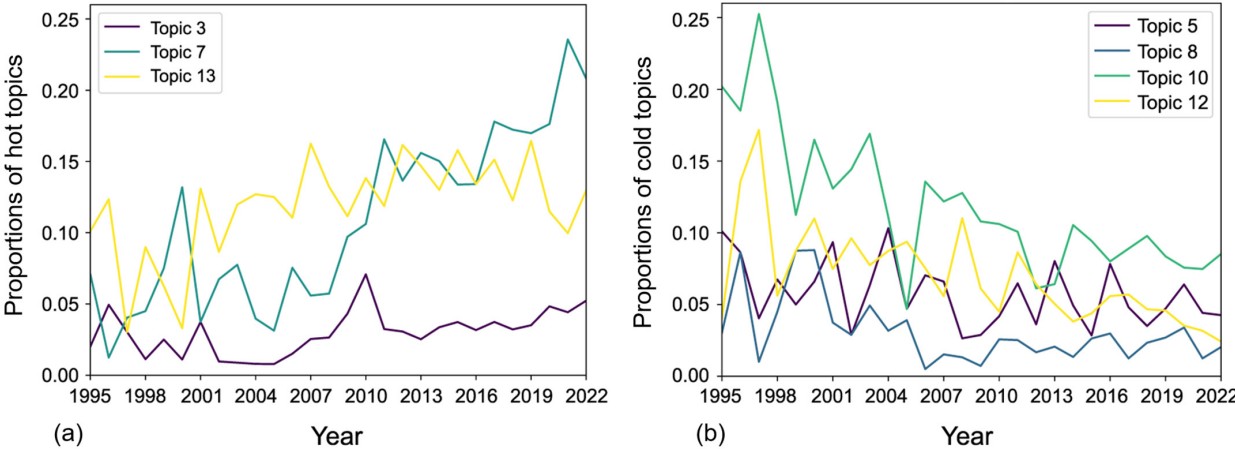

**Figure 5.** Topic proportions of (**a**) hot topics that increase over time and (**b**) cold topics that decrease over time from 1995 to 2022.

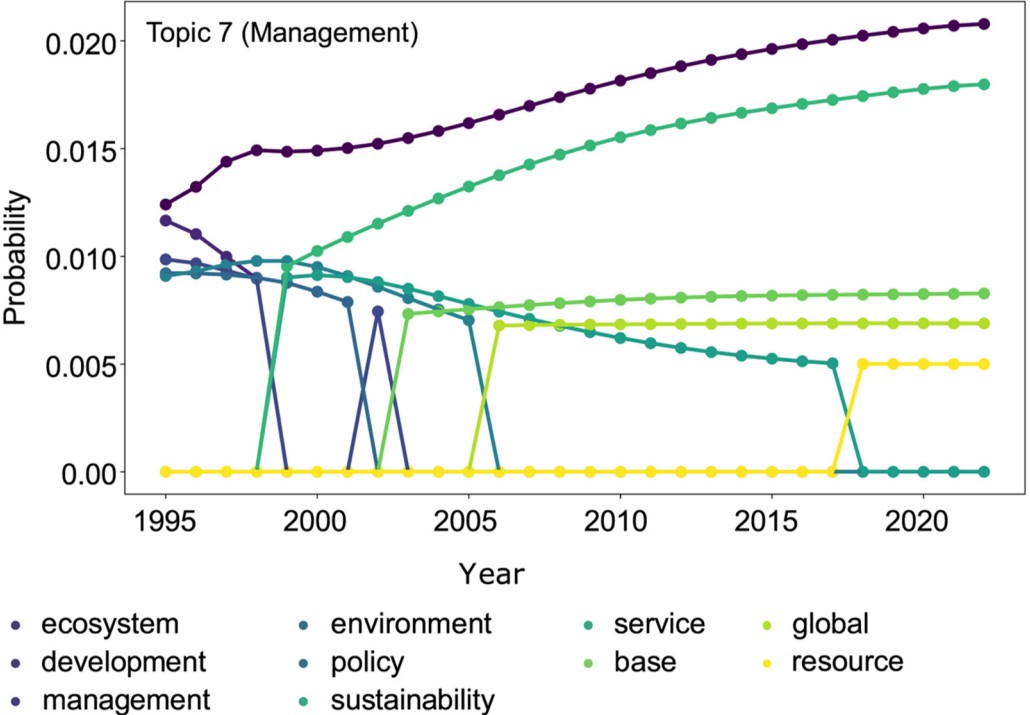

**Figure 6.** Temporal changes of keyword occurrence probabilities in Topic 7 (management) of peatland research studies between 1995 and 2022.

Second, the temporal change in keyword occurrence probability in Topic 8 (deposit) was relatively static (Figure 7). The probability of the appearance of *coal* continuously reduced with time; however, it was still maintained within the ranking. *Deposit*, *alkan*, *basin*, and *plant* did not show significant change within the ranking. *Peat*, *isotop(e)*, and *seam* disappeared from the ranking after 1996, 1999, and 1995, respectively.

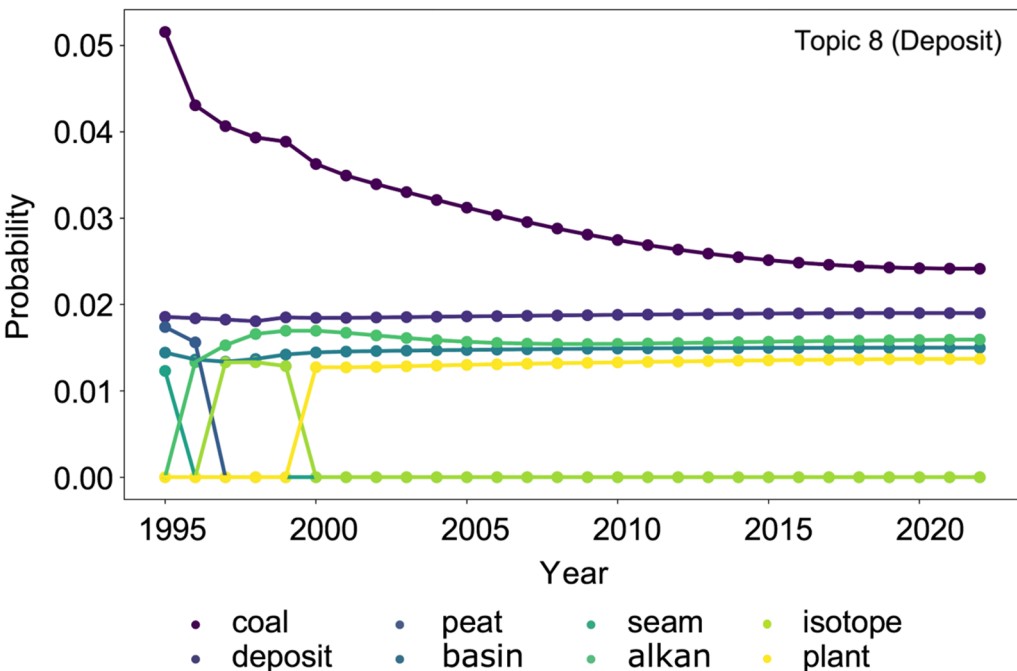

**Figure 7.** Temporal changes of keyword occurrence probabilities in Topic 8 (deposit) of peatland research studies between 1995 and 2022.

### 3.2. Network Analysis

Network analysis showed that Topic 4 had the highest degree, closeness, and eigenvector centralities among the 16 topics (Table 3). Topics 5 (habitat), 9 (fire), and 13 (past environmental change) also followed with overall high centrality. This result can be interpreted as topics 4, 5, and 9 having numerous direct links with other topics (i.e., high degree centrality). The total sum of the distance of the links that connect other topics was shortest in Topics 4, 5, and 13, contributing to the fast transfer of information over the entire network (i.e., high closeness centrality). Additionally, Topics 4, 9, and 13 not only had many links but also high centralities of the linked topics, indicating that they had the greatest impact on the whole network structure (i.e., high eigenvector centrality, as shown in Figure 8). In contrast, Topics 3 (production) and 15 (metal) showed the lowest degree, closeness, and eigenvector centralities, suggesting that they were independently studied rather than closely linked to other topics.

**Table 3.** Degree centrality, closeness centrality, and eigenvector centrality of the 16 topics of peatland research.

| Topic | Degree Centrality | Closeness Centrality | Eigenvector Centrality |
|---|---|---|---|
| 4 | 0.467 | 0.625 | 0.362 |
| 9 | 0.467 | 0.556 | 0.324 |
| 13 | 0.333 | 0.575 | 0.289 |
| 12 | 0.333 | 0.556 | 0.288 |
| 5 | 0.453 | 0.577 | 0.287 |
| 16 | 0.400 | 0.575 | 0.283 |
| 11 | 0.333 | 0.556 | 0.277 |

**Table 3.** *Cont.*

| Topic | Degree Centrality | Closeness Centrality | Eigenvector Centrality |
|---|---|---|---|
| 14 | 0.333 | 0.556 | 0.275 |
| 8 | 0.333 | 0.536 | 0.267 |
| 6 | 0.267 | 0.517 | 0.228 |
| 1 | 0.267 | 0.556 | 0.208 |
| 10 | 0.267 | 0.500 | 0.207 |
| 7 | 0.267 | 0.536 | 0.197 |
| 2 | 0.267 | 0.484 | 0.155 |
| 15 | 0.134 | 0.395 | 0.103 |
| 3 | 0.067 | 0.333 | 0.030 |

Topics are arranged in descending order of the eigenvector centrality score.

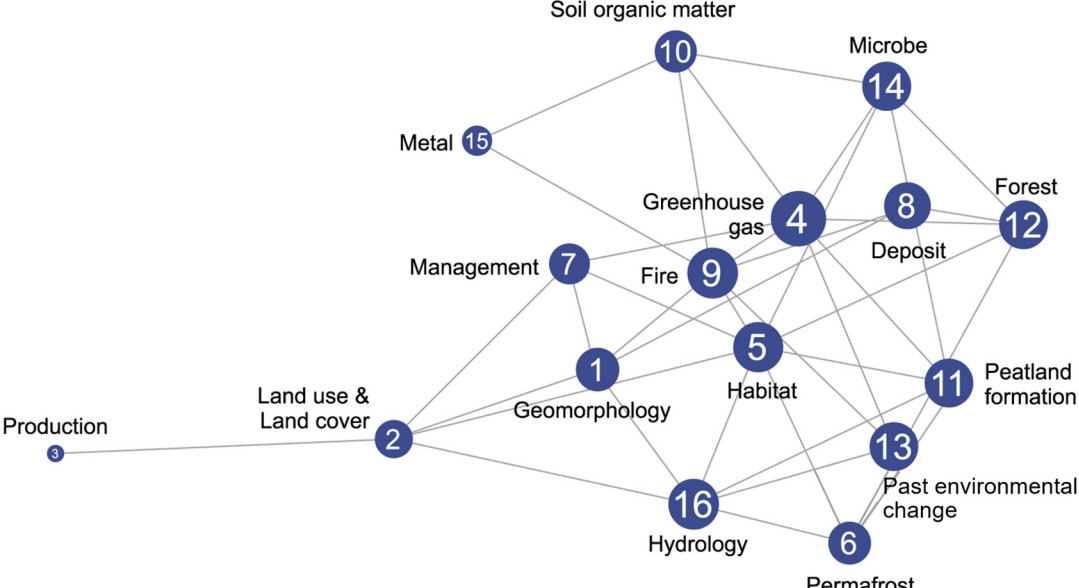

**Figure 8.** Network analysis of the 16 topics of peatland research. The size of a node (i.e., circle) represents the value of eigenvector centrality, and the link (i.e., line) represents similarity among topics calculated using the term frequency-inverse document frequency (TF-IDF) matrix and cosine similarity. The node sizes increase as the value of eigenvector centrality of each topic increases. The link length shortens as the cosine similarity among topics increases. Numerals inside each node indicate the topic number.

## 4. Discussion

### 4.1. Toward Sustainable Management of Peatland under Climate Change

To understand the global trends in peatland research, this study performed DTM and network analysis using text data extracted from titles and abstracts of papers published from 1995 to 2022, searching for using the keyword 'peatland' in ScienceDirect. As a result of classifying 16 topics with DTM, hot topics with significant increases in the topic proportions over time were identified. These hot topics were Topics 7 (management), 13 (past environmental change), and 3 (production), which were mainly related to the management and utilization of peatlands. In contrast, cold topics, with a significant decrease in the topic proportions over time, were Topics 10 (soil organic matter), 12 (forest), 8 (deposit), and 5 (habitat), which were mainly related to the fundamental biological, physical, and chemical characteristics of peatlands. Thus, it can be interpreted that the recent trend is more focused on applied research relevant to the use and management of peatlands in a sustainable way rather than fundamental research focused on biology, physics, and chemistry. This result is in the same context as the results derived by Van Bellen and Larivière [24]. They reported that keywords such as climate change, biodiversity,

management, and restoration significantly increased with time in peatland research, which reflected the increase in studies dealing with problem solving and interactions between humans and the environment.

In particular, Topic 7 (management) accounted for the largest ratio of the number of studies among the total number of studies, and the increase in the rate of topic proportion was also the sharpest. These results suggest that research on the sustainable management of trade-offs and synergy between various ecosystem services in peatlands has recently gained more interest. Topic 7 includes studies that not only consider peatlands as a target of preservation and development but also regard them as a response measure to climate change, such as a means for greenhouse gas reduction. For example, Puspitaloka et al. [41] proposed the restoration of peatlands as a cost-effective measure of conserving biodiversity and reducing greenhouse gases. They emphasized that peatland restoration could be a means to offset carbon emissions to achieve the nationally determined contributions (NDC) to the Paris Agreement, which is a target for greenhouse gas reduction and adaptation voluntarily set by each nation. Bonn et al. [42] pointed out that peatland restoration is eligible for recognition as a means of investment for alleviating greenhouse gas emissions in the carbon market and presented a method of utilizing peatlands for greenhouse gas reduction. Ziegler et al. [43] introduced the usefulness of paludiculture, which involves planting crops while peatlands are rewetted, to prevent drainage and greenhouse gas emission from the peatland. These trends toward sustainable management of peatlands under climate change can be found at national or sub-national levels. For example, Western countries [44,45] and Southeast Asia tropical countries such as Indonesia and Malaysia recently designed a national-level peatland management plan to mitigate climate change [46,47]. Our results also support the view that peatlands are changing over time toward a sustainable management under climate change. Thus, it is necessary to promote integrated and sustainable peatland management at the national and sub-national levels, which includes the preservation and development of various ecosystem services in peatlands and the expansion of reduction to mitigate climate change.

As such, peatlands are increasingly recognized in academia as a means to respond to climate change. In terms of policies, as the Kyoto Protocol adopted detailed execution rules (Decision 2/CMP7) of land use, land-use change, and forestry to be applied during the second commitment in the 16th Conference of the Parties, wetland drainage and rewetting were added as a greenhouse gas reduction method, preparing the basis for the policy. Nevertheless, less than 15% of nations containing peatlands include peatlands in their means of greenhouse gas reduction for their NDC due to the prioritizing goals of development and food security above climate mitigation, a lack of incentives for farmers to improve management practices, and the difficulty of accurate monitoring [48,49]. In this regard, the practical use of peatland as a means of greenhouse gas reduction falls short of expectations. Therefore, a political, technological, and scientific foundation must be prepared by developing national emission factors related to peatlands and models for predicting long-term greenhouse gas reduction so that peatlands can be recognized and utilized as effective reduction methods for NDCs.

*4.2. Dynamic Change in Keywords in Topics with a Large Number of Studies and Increasing Trends*

This present study also analyzed the temporal change in keyword occurrence probability within topics. The results suggested that Topic 7 (management), with the largest topic frequency and the sharpest increase in topic proportion over time, had relatively frequent appearances, disappearances, and changes in the appearance probability of keywords within the top five ranked keywords. This was attributed to the reflection of the relatively dynamically evolving change in research interest within the topic. Despite the frequent changes, there were noticeable keywords such as *ecosystem* and *servic(e)* that consistently maintained their high ranking together, suggesting that the ecosystem service has recently gained more interest in peatland management. On the other hand, for Topic 8 (deposit),

which showed one of the lowest topic frequencies and the weakest increase in topic proportion over time, the appearance, disappearance, and change probability of keywords within the ranking were relatively stagnant, indicating that the temporal change in research interest was not significant and similar research was continuously conducted. For example, the probabilities of the appearance of *coal*, *deposit*, *alkan*, *basin*, and *plant* were continuously maintained within the ranking, indicating that they were consistently interesting themes in ancient peatland deposits. Topic 13 (past environmental change), another topic with a large topic frequency and a sharp increase in topic proportion, showed frequent changes in the probability of keyword appearance, similar to Topic 7, while Topic 15 (metal), with a low topic frequency and weak increase in topic proportion, showed limited changes of probability in keywords, similar to Topic 8. Therefore, our result showed the more frequently topics are studied, the more various interests are developed dynamically over time, which was opposite to that of Gao et al. [50], showing that the popularity of a topic and the evolving dynamics of a topic are unrelated.

### 4.3. Greenhouse Gas Had the Greatest Impact on Other Topics in Peatland Research

Lastly, network analysis was performed to identify the structural relationship between the topics. The most important topic in terms of degree, closeness, and eigenvector centralities was Topic 4 (greenhouse gas). This result implies that Topic 4 had the most direct links with other topics, disseminated information to the entire network at the fastest rate, and was related to other topics with high impact [35,51]. Given the characteristics of peatlands to accumulate peat that stores carbon and emits greenhouse gases when exposed to air, it was considered that greenhouse gas was commonly covered in most studies as the research subject. This result implies that managing and studying peatlands to respond to climate change and to reduce greenhouse gases could promote comprehensive research because the topic related to greenhouse gases significantly impacted other topics in the network. Following Topic 4, Topic 9 (fire) and Topic 13 (past environmental change) also had significant impacts on the network structure in that Topic 9 included research on greenhouse gases emitted from fire in peatland (e.g., Shi et al. [52]), and Topic 13 included research that tracks past climate change based on greenhouse gas data recorded in peatland deposits (e.g., Hong et al. [53] and Zhang et al. [54]).

### 4.4. Limitations of This Study

This present study identified which topics recently attracted more attention in peatland research, to what extent the keywords within the topics changed dynamically, and in what structure the topics were linked to each other. While the previously analyzed trends in peatland research simply focused on which individual keywords were appearing at the time [24], this study used improved tools in that it classified topics based on the groups of keywords and tracked the temporal changes of the topics using DTM and examined structural relationships among topics using network analysis. Since there is a limitation that the details of each topic can only be inferred by relying on the knowledge of the researchers, cross-examination by several researchers with abundant theoretical knowledge and practical experience is required to overcome this limitation, as was conducted in this study. Additionally, the results may be sensitive to the hyperparameters; therefore, it is essential to adjust the hyperparameters to obtain reliable results [31], as was conducted in this study. For further analysis, if qualitative research such as content analysis can be conducted in parallel to explain the changes to each topic in detail, it will be possible to obtain a deeper understanding of the trends of peatland research.

## 5. Conclusions

This study analyzed the temporal change in global trends of peatland research via DTM and network analysis. Based on the results of DTM, we found that the recent trend of peatland research is focused on sustainable management under climate change rather than on its fundamental biological, physical, and chemical characteristics. The more frequently

the topics are studied, the more various interests are developed dynamically over time. In network analysis, the impact of topics related to greenhouse gases on other topics in peatland research was the greatest; therefore, if peatlands are studied in response to climate change, it is expected that overall peatland research will be promoted. However, a gap between academia and policies was found in the field of managing peatlands as a means of responding to climate change, such as Nationally Determined Contributions (NDC), which supported the necessity of effective policies, research, and technology for the use of peatlands as a means of greenhouse gas reduction. This gap should be filled with follow-up studies. Finally, DTM and network analysis are useful tools providing insights about the global trends of peatland research toward sustainable management of peatland under climate change and finding a gap that needs to be dealt with in the near future.

**Supplementary Materials:** The following supporting information can be downloaded at: https://www.mdpi.com/article/10.3390/f14091818/s1, Table S1: List of stop-words for data pre-processing in dynamic topic modeling, Table S2: Representative papers in peatland research for each topic classified by dynamic topic modeling. References of Table S2 were cited from [18,53–213].

**Author Contributions:** Conceptualization, H.Y. and E.C.; methodology, H.Y. and J.C.; software, J.C.; validation, H.Y., J.C., A.-R.Y., R.A.S. and E.C.; investigation, H.Y. and J.C.; resources, A.-R.Y. and E.C.; data curation, J.C.; writing—original draft preparation, H.Y., J.C., A.-R.Y., R.A.S. and E.C.; writing—review and editing, H.Y., J.C., A.-R.Y., R.A.S. and E.C.; visualization, H.Y. and J.C.; supervision, E.C.; project administration, E.C.; funding acquisition, E.C. All authors have read and agreed to the published version of the manuscript.

**Funding:** This study was funded by the National Institute of Forest Science, grant number No. FM0800-2021-03-2023.

**Data Availability Statement:** Publicly available datasets were analyzed in this study. This data can be found here: https://www.sciencedirect.com/ (accessed on 4 May 2023).

**Acknowledgments:** This study was carried out at the National Institute of Forest Science, Republic of Korea. The authors feel gratitude to Himlal Baral for comments on the contents and to Keonju Na for technical support.

**Conflicts of Interest:** The authors declare no conflict of interest.

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
