# Peer review of "Trends of Peatland Research Based on Topic Modeling: Toward Sustainable Management under Climate Change"

_forests, doi:10.3390/f14091818_

Round 1
Reviewer 1 Report
This is a comprehensive survey of peatlands research. The method is sound and results were presented clearly.
There's just some very minor English editing needed. (example: in abstract line 21, "necessity of for policies")
Reviewer 2 Report
General comments
-This manuscript, by authors, studied “
“Trends of Peatland Research Based on Dynamic Topic Model-2 ing and Network Analysis”.
Overall, the topic is not directly of interest to Forests, readers. However, the following are the specific comments on the article concerns, before publication as major revision.
-Figures 1 and 2 need to improve with more details and make it more attractive.
- It is suggested to focus your study from 2000 to 2023 as a recent study and important to all readers. It can show importance in some sentences as an earlier important topic.
Specific Comments and Suggestions
-Title
-Can improve with significant findings
-Abstract
-Can improve this sentence to make it clear.
-Please mention what are significant results.
- “Among these, sustainable management of peatlands has recently gained more interest.” How? give details.
-Significance of your study? Specific findings?
- “This study demonstrates that DTM and network analysis are useful tools for understanding global trends of peatland research”. For what?
-Introduction
-Add more details and importance of wetlands, peat lands, and related studies done. Need to summarize and be specific with your concerned study.
-Can add more recent references. You may consider the following or more relevant for nutritious values and gases emission.
“Application of wetland plant-based vermicomposts as an organic amendment with high nutritious value”
-Must revise your introduction section to show the importance and globally significant research and how you are extracting your own study. Be specific.
-Materials and Methods
-Just on the basis of the word “Peat or peat land” is not enough. Must clear your objectives of study.
-How you selected 16 topics? On which basis?
-Methodology also needs to summarize and be specific. Make a table to show your doses with treatments clearly.
-Table 1 is not needed.
-publication period is too long and difficult to present actual details. Make it shorter with updated studies.
- Why need this table? “Table S1. List of stop-words for data pre-processing in dynamic topic modeling”
-Results
-In Figure 3, from 1950 -1980 shows negligible effects. It is suggested to focus your study from 2000 to 2023 as a recent study and importance to all readers.
- In Table S2, Give topics instead of Topics 1, 2,,…..?
- In Fig 8, How do you explain linkage? Not clear.
SEM model figure? How?
Discussion
- Discussion is also weak. Add more recent references?
-Please add other study references to support your current study and its significance.
-For specific discussion can also use sub-headings to show your significant results.
Conclusions
-Not necessarily how many papers were published. Why does it need to be published?
-Significant results and importance showing problems and solutions.
- “Through this study, it was confirmed that DTM and network analysis are useful tools providing insights about the 416 global trends of peatland research.” How?
Reviewer 3 Report
The article ,, Trends of Peatland Research Based on Dynamic Topic Model- 2ing and Network Analysis" is a summary of a review of peatland research. From a scientific point of view, in my opinion whether it adds anything new to science, rather not. Appreciating the contribution and labour of the authors, I think it is worth publishing.
Reviewer 4 Report
Comments on the manuscript (forests-2532115) entitled 'Trends of Peatland Research Based on Dynamic Topic Modeling and Network Analysis'
It is an interesting manuscript presenting the trend of peatland research, using the methods of dynamic topic modeling and network analysis. The results have important implications for peatland research. Presently, I have a major concern on how to select the 16 topics and 10 keywords of each topic. Statistical analyses can produce their results, but the rationality should be assessed very carefully.
In Table 2,
Topic 1 'Geomorphology' is fine, but some keywords are missing, such as microtopography, hollow, hummock
Topic 8 'deposit' mainly mentions paleo peat sedimentation. It is fine, but overlapped with Topic 1 'Geomorphology' and Topic 13 'Climate record'
Topic 10 'soil organic matter' can be 'carbon cycling', some keywords are missing, such as dissolved organic carbon, carbon burial,
Topic 12 'Forest' can be replaced by 'Plant', some dominant plants are missing, such as Sphagnum, inter-species competition
Topic 13 'Climate record' can be 'past environmental changes', some keywords are missing, such as pollen, plant macrofossil, testate amoeba, Humification. I suggest that 'China' can be removed. In this topic, some representative papers are lake papers rather than peatland paper, in Topic 13, such as Zhang et al., 2022, Ning et al., 2017. This issue might also exist in other topics.
If the selectivity of topics and keywords are not rational, the results are not reliable. Therefore, I strongly suggest that the topics and keywords should be assessed very carefully.
English writing if fine for me.
Round 2
Reviewer 2 Report
The authors made great efforts to improve the manuscript.
Reviewer 4 Report
Authors have revised the manuscript based on previous comments and suggestions. However, further revisions are needed to improve the manuscript.
Firstly, It is confused that the study timespan is different in the Methods section, 11519 papers since 2022 in L132, 10158 papers from 1953 to 2022. Please check it.
Secondly, some representative papers in Table S2 are not related to peatlands!!! please check them very carefully. For example, Topic 13, Zhang et al., 2022, Yang et al., 2021, Ning et al., 2017 are lake papers, not peatland paper!!! Similar problems also exist in other topics.
Thirdly, English writing should be improved. Minor revisions are listed as follows.
1) L30: using 'peatlands'
2) L60: can also 'store' water by ...
3) L61-62: 'peatlands' can store nitrogen[13]. Moreover, they also have 'an' important...
4)L76-77: is not only 'a' serious.... but also 'a' cause of conflicts...
5) against the 'crises'
6) L110 and other places 'between topics' replaced by 'among topics'
7) L113 and L510: 'in the' near future
8) L138: global trends of peatland research
9) L139 'the manuscript' replaced by 'the papers'
10) L167 was 'the' highest
11) L198, L199 'according to year' replaced by 'with time'
12) L264: please specify 'in ancient peatlands'
13) L268: other 'organic' matter
14) L274: remove 'past' between 'the' and 'Holocene'
15) L395: consider 'peatlands' as
16) L396: regard 'them' as
17) L424 and the 'difficulty' of
18) L446 in 'ancient' peatland deposits
English writing should be improved.
